

# Trends in normalized difference vegetation index (NDVI) associated with urban development in northern West Siberia

**Igor Esau[1], Victoria V. Miles[1], Richard Davy[1], Martin W. Miles[2], Anna Kurchatova[3]**

[1]{Nansen Environmental and Remote Sensing Centre / Bjerknes Centre for Climate Research, Bergen, Norway}

[2]{Uni Research Climate / Bjerknes Centre for Climate Research, Bergen, Norway / Institute of Arctic and Alpine Research, University of Colorado, Boulder, USA}

[3]{Institute of the Earth's Cryosphere / Tyumen Oil and Gas University, Tyumen, Russia}

Correspondence to: I. Esau (igor.ezau@nersc.no)

## Abstract

Exploration and exploitation of oil and gas reserves of northern West Siberia has promoted rapid industrialization and urban development in the region. This development leaves significant footprints on the sensitive northern environment, which is already stressed by the global warming. This study reports the region-wide changes in the vegetation cover as well as the corresponding changes in and around 28 selected urbanized areas. The study utilizes the normalized difference vegetation index (NDVI) from high-resolution (250 m) MODIS data acquired for summer months (June through August) during 15 years (2000–2014). The results reveal the increase of NDVI (or "greening") over the northern (tundra and tundra-forest) part of the region. Simultaneously, the southern, forested part shows the widespread decrease of NDVI (or "browning"). These region-wide patterns are, however, highly fragmented. The statistically significant NDVI trends occupy only a small fraction of the region. Urbanization destroys the vegetation cover within the developed areas and at about 5–10 km distance around them. The studied urbanized areas have the NDVI values by 15% to 45% lower than the corresponding areas at 20–40 km distance. The largest NDVI reduction is typical for the newly developed areas, whereas the older areas show recovery of the vegetation cover. The study reveals a robust indication of the accelerated greening near the older urban areas. Many Siberian



cities become greener even against the wider browning trends at their background. Literature
discussion suggests that the observed urban greening could be associated not only with special
tending of the within-city green areas but also with the urban heat islands and succession of
more productive shrub and tree species growing on warmer sandy soils.

## 6    1    Introduction

Significant shifts in the vegetation land cover and biological productivity manifest rapid climate
change in the northern high latitudes (Hinzman et al., 2005; Groisman and Gutman, 2013). As
in other polar regions, biomes of northern West Siberia (Fig. 1), hereafter referred to as NWS,
respond on global warming with complex patterns of multidirectional changes (Bunn and
Goetz, 2006; Lloyd and Bunn, 2007; McDonald et al., 2008; Walker et al., 2009; Bhatt et al.,
2013). Although the patterns are geographically fragmented (Elsakov and Teljatnikov, 2013;
Barichivich et al., 2014; Guay et al., 2014), the northern biomes (tundra and forest-tundra)
demonstrate widespread increase of the vegetation productivity ("greening"). Its attribution to
tall shrub and gramminoids in tundra ecosystems (Frost and Epstein, 2014) and enhanced tree
growth in forest-tundra ecosystems (Urban et al., 2014) suggested that we observe transitions
to alternative, potentially more productive ecosystems rather than anomalouesly enhanced
biological productivity in response to global warming (Kumpula et al., 2012; Macias-Fauria et
al., 2012). These enviromental shifts have been unfolding for at least the last three decades
(Keeling et al., 1996; Myneni et al., 1997) albeit at the slower pace since 2003 (Bhatt et al.,

21    2013).

22         Simultaneously, the southern biomes (northern and middle taiga forest to the south of

about 65°N) demonstrate decrease of productivity ("browning"). This browning has been
robustly associated with decreasing mass of the green leaves, which has been linked to Siberian
forest acclimation without the transition to an alternative ecosystem (Lapenis et al., 2005; Lloyd
and Bunn 2007). Although the grown trees are stressed by increasing temperatures, other types
of the vegetation cover (grass- and shrub-lands, mires) and disturbed patches with young tree
seedlings exhibit greening within the same bioclimatic zone.

29         The persistent greening trends of the disturbed vegetation cover is in focus of this study.

Over the last 30–50 years, exploration of oil and gas reserves promoted rapid industrialization
and urban development of NWS. It makes this part of the circumpolar region particularly
interesting for studies of the emerging alternative ecosystems. The development has left



significant footprints on the vegetation cover. These footprints were found not only at the
development sites but also across large distances (Walker et al., 2009; Kumpula et al., 2012).
More specifically, Kumpula et al. (2011) reported extensive transformation from shrub- to more
productive (greener) grass- and sedge-dominated tundra in NWS that reclaim artificial terrain
disturbances at Bovanenkovskiy gas field, Yamal peninsula (70°N). Similar environmental
shifts towards more productive plant successions on disturbed patches were found in studies of
post-mined sand and sandy loam quarries around Noyabrsk at 63°N (Koronatova & Milyaeva,
2011). This transformation across open woodlands and forest-tundra was found to be more
ambigues. There has been found 26% (from 2316 to 1715 g $m^{-2}$) decrease of the total
aboveground biomass due to enhanced bog formation on the background of widespread
greening of better drained sandy patches (Moskalenko, 2013). Detailed study by Sorokina
(2013) revealed the increased soil temperature by 1K to 4K on the disturbed patches within
forest, mire and tundra ecosystems. The reclaiming vegetation cover is characterized by
increasing share of shrubs (from 18.5% to 27.8% in the forest ecosystems) and grasses (from
less than 0.1% to 8.8%) in its above-ground biomass.
Although disturbed patches occupy a relatively small fraction in NWS, their
contribution to the observed vegetation productivity changes can be disproportionally
significant. For example, for the Yamal peninsula, Elsakov and Teljatnikov (2013) concluded
that its widely referred greening (Macias-Fauria et al., 2012) could be traced to statistically
significant changes over just 4.8% of the total area. Moreover, this greening was linked to shrub
growth on the patches with disturbed permafrost or within the patches of already establish
shrubs (Walker et al., 2009). Barichivich et al. (2014) noted that browning in southern forest
biomes could be also partially associated with disturbances.
The reviewed literature broadly agrees that the observed transformation of vegetation
cover represents an alternative ecosystem state (Kumpula et al., 2012; Macias-Fauria et al.,
2012) and a new geoecological regime (Raynolds et al., 2014) acclimated to higher
temperatures that are likely to persist. This point of view received strong support from the plot-
scale International Arctic Tundra Experiment, which simulated warmer climate conditions in
high latitudes (Walker et al., 2006; Elmendorf et al., 2012). The question remains whether the
plot-scale experiments and field studies at a few carefully selected locations, e.g., Vaskiny
Dachi in the central Yamal peninsula (Leibman et al., 2015), could be extrapolated for
assessment of the larger scale transition to alternative ecosystems. To approach this question,



we use an opportunity provided by the above-mentioned intensive urbanization in NWS. The
urbanization created large number of relatively large-scale (tens of km across) artificial
disturbances across all major regional biomes. This study looks at systematic differences in the
apparent vegetation productivity trends between the undisturbed background and the areas
disturbed by urbanization and industrial development.

6       As in several previously published studies, this study utilizes a Normalized Difference

Vegetation Index (NDVI) derived from satellite data products from the Moderate Resolution
Imaging Spectroradiometer (MODIS) onboard of the Earth Observing System-Terra platform
satellite. Studies by Frey and Smith (2007), Epstein et al. (2012), Macias-Fauria et al. (2012)
and others demonstrated that MODIS NDVI could be successfully used in vegetation
productivity assessment for Siberian biomes. This study details the previously reported NDVI
maps of the region and expands them over the last 15 years (2000–2014) period. The vegetation
productivity of emerging alternative ecosystems are calculated at and around of 28 urbanized
areas, hereafter referred to as cities. These 28 cities are located in four northern biomes ranging
from tundra (Bovanenkovo) to middle taiga forest (Uray).

16       This paper has the following structure: Data and methods of the study are described in

Section 2. Section 3 presents the analysis and results for both the background changes in NDVI
and NDVI in and around the selected cities. Section 4 discusses the results in the context on
previous research and emphasize the role of the cities. Section 5 outlines conclusions.

## 2   Data and methods

This study is based on analysis of the maximum summer NDVI, which we denote as *NDVImax*.
*NDVImax* was obtained from MODIS NDVI 16-day composites with the 250-m spatial
resolution (MOD13Q1). The data were downloaded from the NASA's Earth Observing System
Data and Information System (EOSDIS) for 15 summers (June–July–August, JJA) covering the
period 2000–2014.

### 2.1   The maximum Normalized Difference Vegetation Index, *NDVImax*

NDVI is defined as a normalized ratio of reflectance factors in the near infrared (NIR) and red
spectral radiation bands

$$NDVI = \frac{\rho_{NIR} - \rho_{RED}}{\rho_{NIR} + \rho_{RED}} \tag{1}$$



where $\rho_{NIR}$ and $\rho_{RED}$ are the surface bidirectional reflectance factors for their respective
MODIS bands on the TERRA platform satellite. NDVI exploits the contrast between the red
and NIR reflectance of vegetation, as chlorophyll is a strong absorber of the red light, while the
internal structure of leaves reflects highly in the NIR. The greater the difference between the
reflectance in the red and NIR portions of the spectrum, the more chlorophyll is found in
vegetation canopy. Vegetation generally yields positive NDVI values, which approach +1 with
increasing plant chlorophyll content or green aboveground biomass. NDVI with the values
below 0.2 generally corresponds to non-vegetated surfaces, whereas green vegetation canopies
have NDVI greater than 0.3.
NDVI in remote sensing studies is a popular proxy for gross photosynthesis, and
therefore, for vegetation productivity (Goetz et al., 2005). A strength of NDVI is its
normalization, which makes it relatively insensitive to radiometric attenuation (e.g., by cloud
shadows) present in multiple bands. The main weakness of NDVI is its inherent nonlinearity
that leads to asymptotic saturation of NDVI over higher biomass conditions. This saturation,
also known as the NDVI degradation, is particularly strong in the areas with higher canopy
background brightness corresponding to the most productive biomes. By contrast, the biomass
accumulation in the less productive biomes could be approximated with a linear regression
model. NDVI typically does not degrade in tundra (Raynolds et al., 2012), but the degradation
should be empirically taken into account in NDVI interpretations for more southern biomes
(D'Arrigo et al., 2000; Zhang et al., 2004).
A remarkably strong correlation ($R^2 = 0.94$, $p<0.001$) was found between total above-
ground phytomass sampled at the peak of summer and the maximum annual NDVI (*NDVImax*)
in studies of the North America and Eurasia transects (Raynolds et al., 2012). This strong
relationship encouraged us to use *NDVImax* in this study. *NDVImax* is a more conservative
characteristic of the vegetation cover that is linked to the total biomass at the late phenological
phases. Moreover, *NDVImax* eliminates seasonal variations and the relative shifts between
phenological phases in different climatic zones. Hence, *NDVImax* is particularly convenient for
and frequently used in the environmental studies dealing with long-term and large-scale
changes, including effects of the climate change.
*NDVImax* was obtained from MOD13Q1 data product. This product is distributed in
adjacent non-overlapping tiles with the side of approximately 10 degrees (at the equator) and
the Sinusoidal (SIN) tile grid projection (Solano et. al, 2010). Five tiles to cover the entire area





of interest (total 20 tiles per each summer) were downloaded and imported into the ArcGIS
geographic information system. Images were combined and re-projected from the original SIN
to the Universal Transverse Mercator projection (UTM Zone 42N, WGS84 ellipsoid). The data
were quality-filtered by the MODIS reliability data provided together with the MOD13Q1
product. Only data of the highest quality, which excluded snow/ice- and cloud covered pixels,
were retained. The NDVI > 0.3 criterion was used here to exclude water, bare soil and other
non-vegetated pixels from the analysis. The data gaps in the raster mosaic pixels were then
filled with information using the nearest neighbor statistical interpolation from the surrounding
pixels with data. Finally, *NDVImax* maps for each summer were obtained through identification
of the maximum NDVI value from each 16-day composite for each pixel. The analysis operates
with these *NDVImax* maps of the 250-m resolution covering the 15-year period 2000–2014.
## 2.2 *NDVImax* in and around selected cities
The NWS territory provides a unique opportunity for statistical study, which compares the
effect of recent urbanization along with the effect on the climate change on the vegetation land
cover in and around cities across several northern biomes. Twenty-eight (28) cities, some of
them with more than 100,000 inhabitants (see Table 1), were selected. *NDVImax* was studied
within 40 km buffer zones around each city. Each buffer zone was broken into 8 rings of 5 km
width centered at the city core zone. This approach is similar to one used by Zhang et al. (2004).
Let us introduce $\langle NDVImax(t, i, n) \rangle$ as the *NDVImax* value in the year $t = 1 \dots 15$ for
the ring number $i = 0 \dots 8$ (the ring $i = 0$ corresponds to the city core and $i = 8 -$ to the most
distant background ring) in the city $n = 1 \dots 28$ that is averaged over all pixels within the ring.
This approach does not differentiate between the vegetation productivity changes and the
changes of the vegetation biological composition (succession). In this sense, it will be
insensitive to the *NDVImax* trends due to urban expansion, shifting vegetation disturbances and
re-vegetation. As the $\langle NDVImax(t, i, n) \rangle$ values strongly vary from city to city and across
different biomes, it is convenient to introduce a relative footprint of a city as

$$F(t, i, n) = \frac{\langle NDVImax(t,i,n) \rangle}{\overline{\langle NDVImax(\iota,n) \rangle}} \qquad (2)$$

and the relative linear trends, $R(i, n)$, which are computed by the least-squares fit of $F(t, i, n)$
to the first-order polynomial for the ring $i$ in the city $n$, $\overline{\langle NDVImax(\iota, n) \rangle}$ is time-averaged
$\langle NDVImax(t, i, n) \rangle$. Because the study operates with rather short 15-year time series, the years




with the maximum and the minimum *NDVImax* can strongly impact the trends. These two years
(2002 and 2014 for the majority of cities) can be considered as outliers as their *NDVImax* were
beyond three standard deviations of the respective time series. We demonstrate this for four
cities (Bovanenkovsky, Nadym, Noyabrsk and Surgut) in Fig. 3. Therefore, the years with
minimum and maximum *NDVImax* were excluded from the trend fitting. We will also consider
the differences in the relative footprints, $\Delta F(i,j,n) = \overline{F(t,\iota,n)} - \overline{F(t,j,n)}$, and relative
differences in the trends, $\Delta R(i,j) = R(i,n) - R(j,n)$, between the rings $i$ and $j$ for the city $n$.
The urbanization footprint could be characterized through the divergent trends: $\Delta R(0,8)$ –
between the city core and the corresponding natural land cover; $\Delta R(0,5)$ – between the core
and the first ring around it; and $\Delta R(5,8)$ – between supposedly the most affected 5-km ring and
the background 40-km ring.
**3    Results**
**3.1    Regional *NDVImax* patterns and trends**
Our study expands, updates and details the results of the *NDVImax* analysis for 1981–1999 by
Zhou et al. (2001), 1982–2003 by Bunn and Goetz (2006), 1982–2008 by Beck and Goetz
(2011), 1982–2011 by Barichivich et al. (2014) and 2000–2009 by Elsakov and Teljatnikov
(2013). Two novel aspects should be mentioned in this context: (1) Whereas the previous
studies analyzed coarse-resolution data, which is likely to exaggerate the extent and magnitude
of the *NDVImax* trends (Zhao et al., 2009; Elsakov and Teljatnikov, 2013), we use the fine-
resolution (250 m) data; and (2) Fine-resolution data give an opportunity to trace the changes
to specific biomes within the same bioclimatic zone and to reveal effects of urban disturbances.
The updated mean *NDVImax* and *NDVImax* trend maps (2000–2014) are shown in Fig.
1a and 1b, respectively. *NDVImax* in NWS generally decreases from the southwest to the
northeast of the territory. The largest *NDVImax* values (the most productive vegetation) are
found along the Ob river and between the Ob river and Ural mountains, whereas the central,
swamped part of NWS has much lower *NDVImax*. We observe that *NDVImax* is significantly
higher on river terraces with better-drained, sandy soils, which are warmer in summertime and
have deeper seasonal active layer. The new maps confirm continuing widespread greening in
tundra and forest-tundra biomes. However, this greening is highly fragmented and to the large
degree could be associated with sandy soils as well as with smaller greening patches associated
with permafrost destruction, landslides, thermokarst and other local disturbances. Fig. 1b shows




the statistically significant trends. It clearly demonstrates that the previously reported
widespread greening trends are statistically insignificant. The most significant areas of greening
are found in Taz and southern Gydan peninsulas. This finding is in good agreement with
previously reported plot scale studies using LANDSAT images. Table 2 and Fig. 2 aggregate
the greening and browning trends in the NWS biomes. They show that the forest biomes exhibit
more widespread and larger magnitude browning. The maximum area fraction of browning
(21.3%) was found in middle taiga and the minimum area fraction (8.9%) – in tundra biomes.
Contrary, the area fraction of greening is the largest (81.7%) in forest-tundra and the smallest
(35.5%) in middle taiga biomes.
Comparison with Lloyd and Bunn (2007), Bhatt et al. (2013) and Elsakov and
Teljatnikov (2013) studies further reveals that the area of more productive vegetation cover
continue to grow as well as the production in shrub- and graminoid-dominated ecosystems.
Despite colder recent winters (Cohen et al., 2013) and somewhat damped summer warming
(Tang and Leng, 2012), the greening now dominates the changes in all four biomes. We observe
the strongest greening near the southern tundra boundary. This pattern is consistent with
previously noted shrubification and treeline advance in this area (Devi et al., 2008; MacDonald
et al., 2008).
**3.2 *NDVImax* patterns and trends in and around 28 cities**
Disturbances of the vegetation cover around 28 cities in NWS considerably modify the
observed complex pattern of the background *NDVImax* trends. In this analysis, we distinguish
the city core, $i = 0$, with strongly disturbed vegetation cover and therefore low *NDVImax* and
the rings $i = 1 \dots 8$ where the area of disturbances progressively decrease with the distance from
the city core. Fig. 3 shows the analysis of *NDVImax* changes for four typical cities located in
four different biomes. The observed convergence of the statistical properties of *NDVImax* in
the rings $i = 4 \dots 7$ to those in the ring 8 supports the intuitive assumption that the area fraction
of urban disturbances is gradually reduced with the distance from the city. Indeed, the
correlation coefficients between time series of $F(t, 0, n)$ and $F(t, i, n), i = 1 \dots 8$, are
decreasing $i$ (see the upper panels in Fig. 3).
Now, we consider differences between the background *NDVImax* trends and the trends
over the disturbed vegetation cover in and around the cities. Fig. 4a and 4c show that all cities
have strongly reduced *NDVImax* values. Some cities have *NDVImax* reduced by more than 30%



as compared to the background. As it has been already suggested by Fig. 3 (for Nadym and,
particularly, for Noyabrsk), the closest 5-km ring ($i = 1$) often exhibits higher *NDVImax* than
the more distant background. This unusual feature could be traced down to the preferential
location of the cities on generally greener river terraces. So that the greener patches contribute
more heavily to the mean *NDVImax* of the inner rings.
Apart from lower *NDVImax* in the city cores, Fig. 4d reveals two opposite dependences
between the vegetation disturbance and the vegetation productivity. The cities with larger
relative *NDVImax* reduction (the large negative ΔF) demonstrate the accelerated *NDVImax*
recovery (the large positive Δ$R$). This rapid recovery at the initial stages of the vegetation
succession has been repeatedly noted in several plot-scale studies (Sorokina, 2003; Archegova,
2007; Koronatova and Milyaeva, 2011) and high-resolution satellite image analyses (Kornienko
and Yakubson, 2011; Kumpula 2011). Unfortunately, majority of these studies have been
published in Russian only. Intercomparison with Fig. 4c, where this dependence is less
pronounced, suggests that the vegetation cover in cities is more resistant to the stress in the
extreme warm and cold years. Moreover, the protecting effect of the city is stronger for the
northern cities responding with strong greening due to extremely warm summer temperatures.
It is interesting to observe that the effect of cities does not exhibit clear regular
dependence on the city population or specific location. The only visible effect in Fig. 4a could
be attributed to the age of the city – the younger northern cities continue to destroy vegetation
cover as they expand, whereas the vegetation cover in the established southern cities recovers.
Fig 4b shows that the disturbances have the largest positive effect on both the mean *NDVImax*
and its trends in the forest-tundra biome. Here, 6 out of 7 cities induce the accelerated greening
in the 5-km ring.
The biome-averaged impact of the urban disturbances on *NDVImax* is given in Fig. 5.
The analysis show that the city footprint is visible at large distances. Even at the 15 km distance
from the city core, *NDVImax* is systematically higher in the northern biomes and lower in the
middle taiga biome. Moreover, *NDVImax* is also systematically the highest in the closest 5-km
ring where the area of disturbances is the largest. Fig. 6 illustrates the vegetation changes
leading to higher biological productivity of the disturbed land patches in the northern biomes.
The newly established tree seedlings and shrubs, which are more productive relative to the
background in the northern biomes, are less productive relative the background of the mature
trees in the middle taiga biome.





## 4   Discussion

The results of the presented high-resolution *NDVImax* analysis in 2000–2014 show continuing changes in the vegetation land cover and vegetation productivity in NWS. In general, the northern biomes (tundra and forest-tundra north of about 65ºN) and to some degree open swamped areas within the forest biomes demonstrate persistent greening, whereas the forested areas showed similarly persistent browning. More detailed analysis has however revealed that the statistically significant changes are highly fragmented. Such changes occupy only a minor fraction of NWS and often could be collocated with natural or anthropogenic disturbances of the vegetation cover. This observation lead Macias-Fauria et al. (2012) and Kumpula et al. (2012) to hypothesize that those disturbances help to establish alternative, more productive ecosystems of gramminoids, shrubs and tree seedlings. In this discussion, we consider the major climatological and physical factors that may support the alternative vegetation cover.

The cold continental climates of the NWS (the Köppen–Geiger climate types Dfc and ET) with relatively high amount of annual precipitation determine a short vegetation period and strong dependence of the maximum vegetation productivity on summer temperatures (Barichivich et al., 2014). This dependence is not of universal character across biomes. The higher spring–summer temperatures (Bhatt et al., 2013; Ippolitov et al., 2014) and larger amount of accumulated snow (Bulygina et al., 2014) favor the tall deciduous shrub (*Betula nana* and different *Salix species*) in tundra and forest-tundra (e.g Sturm et al., 2001; Elmendorf et al., 2012). Similarly, grasses (graminoids) respond on higher spring temperatures with higher biomass production. It should be emphasized however that the associations between the *NDVImax* trends and productivity of the specific ecosystems within the arctic biomes remain rather loose (Frey and Smith, 2007; Kornienko and Yakubson, 2011). Moreover, even synthetic *NDVImax* time series (Guay, et al., 2014) are too short and too variable for statistically robust conclusions over the major part of the area.

The response on the observed climate change is different in the forest biomes. Summertime surface air temperatures increased only weakly in central and southern NWS (Ippolitov et al., 2014) likely being damped by increasing cloud cover (Tang and Leng, 2012; Esau et al., 2012). The temperature trends in the winter months were negative (Cohen et al., 2013; Outten and Esau, 2011; Outten et al., 2013). Thus, there were no consistent warming trends over this territory but rather a few summer heat waves (e.g. in 2002, 2007, 2012) with certain impact on the forest productivity.





As the reviewed literature disclose, the direct impact of the climate factors on the
vegetation cannot be established unambiguously. Even conclusions derived from the analysis
of common data sets differ on the relative role of the climate change and the natural decadal
variability. For example, Melnikov et al. (2004) concluded that the climatic trends are only
weakly discernable in the active soil layer data, whereas Streletsky et al. (2012) found a
significant increase (up to 0.3 m) of the active layer thickness. Moreover, simulations of the
forest biomass productivity with a stand-alone dynamic vegetation model by Schumann and
Shugart (2009) produced no significant productivity change at Siberian forest sites for the
warming below 2K. At the same time, the observed biome-wide dichotomy of the grass-covered
area greening versus the tree-covered area browning was reproduced in the anthropogenic
global warming experiments with the Community Atmospheric Model version 3 with the Lund-
Potsdam-Jena dynamic vegetation model (Jeong et al., 2012).
A more coherent picture has been suggested by studies of vegetation reclaiming the
natural or anthropogenic disturbances (Kornienko and Yakubson, 2011; Kumpula, 2011). The
identified, very fragmented pattern of changes and the larger changes near the cities suggest the
large impact of soil and vegetation cover disturbances on of the *NDVImax* trends. The soil
disturbances significantly modify the surface heat balance in the area. Pavlov and Moskalenko
(2002) and more recently Yu et al. (2015) showed that the disturbed soils in tundra and northern
taiga (near Nadym) are warmer and accumulated more heat (thaw deeper) during the summer
seasons. Fig. 6 illustrated that he disturbed soils are better drained, so that the latent heat flux
is reduced. More general discussion however should account for other physical and biological
processes in the active soil layers. Better drainage gives better rooting conditions for forbs and
trees as well as increases organic mass loss in soils, litter decomposition and decreases moss
productivity (Hicks Pries et al., 2013). Yakubson et al. (2012) reported high correlation between
the enhanced *NDVImax* and dryer soils at the disturbed areas around Bovanenkovo. Similar
dependences were found over Taz peninsula (Kornienko and Yakubson, 2011). Moreover,
Brunsel et al. (2011) showed that the boundary layer dynamics might additionally increase the
surface temperature heterogeneity, enhancing the sensible heat flux by as much as 50 Wm$^{-2}$ for
larger disturbances. Our results for the 5-km ring seem to support this feedback hypothesis.
Cities and industrial installations in NWS are frequently built-up on sand beds. It may
partially explain the more positive *NDVImax* trends found in and around the city cores.
Koronatova and Milyaeva (2011) studied plant succession over 1999–2010 in post-mined sandy



and loam quarries around Noyabrsk city. The quarries were colonized by pine seedlings already
during the first 5 years, skipping the grass community stage, and an open pine community with
green moss and lichen species was established by year 20.
Finally, it is fruitful to comment on the relative roles of climate changes and the
anthropogenic disturbances introduced through the new infrastructure development. Yu et al.
(2015) compared the vegetation changes around Nadym using high spatial resolution imagery
acquired in 1968 and 2006. This area corresponds to the buffer zones shown in Fig, 3 for this
city. They concluded that about 9% of the area revealed increase in vegetation cover in response
to climate warming while 10.8% of the area had decrease in vegetation cover due to the
infrastructure development and related factors (logging, tracking etc). The direct mechanical
impact on the vegetation cover was very localized (mostly within 100 m from the infrastructural
objects), but indirect biophysical impacts, such as changes in the surface hydrology, heat
balance and ecosystem damages (e.g. fires) were found over significantly larger areas. These
wider indirect impacts of urbanization are visible not only in highly aggregated analysis in the
present study but also in plot-scale experiments. The global warming favors expansion of wood
vegetation over the permafrost. However, the mechanical removal of shrubs in a Siberian tundra
site initiated permafrost thaw converting the plot into waterlogged depression within five years
(Nauta et al., 2014). Thus, there seems to be a concert between the impacts on vegetation cover
and the soil cryosphere induced by global warming and anthropogenic disturbances.
**5 Conclusions**
This study presented the *NDVImax* analysis of the MODIS NDVI data product MOD13Q1 with
250 m spatial resolution over 2000–2014. We obtained the maps of *NDVImax* and *NDVImax*
trends for the northern West Siberia region where intensive oil and gas exploration created a
large number of anthropogenic disturbances. These new maps were discussed in the context of
the previously published low- and high-resolution NDVI analysis available for this region. The
obtained *NDVImax* trends are highly fragmented, but confirm the observed dichotomy of
northern greening (the increasing vegetation productivity) versus southern browning (the
decreasing productivity). The statistically significant *NDVImax* trends occupy only a small
fraction of NWS. The most significant trends are found on the territories with sandy soils and
with larger concentration of soil disturbances. Thus, the new map substantially corrects the



previous picture of the vegetation cover changes in NWS and suggests stronger resilience of
undisturbed vegetation cover to the climate change on decadal time scales.
It has been proposed that the disturbances might help to establish alternative, more
productive vegetation cover. We used *NDVImax* data in and around 28 regional urbanized and
industrial areas to compare the vegetation productivity of the reclaiming and background plant
communities across all four major biomes. We assumed growing concentration of disturbances
towards the city cores. The results indicated that the reclaiming plant communities tend to be
more productive. Only some cities in middle taiga biome, notably the largest cities of Surgut
and Niznevartovsk on the Ob river, exhibit lower productivity in the closest city proximity.
As it was expected, we found *NDVImax* at the city cores to be 15% to 40% lower than
over the corresponding background. At the same time, many cities has become significantly
greener over the analyzed 15 years. This tendency reflects both the targeted efforts to create
more environmental friendly residential areas and the urban heat island impact on the active
soil layer thickness and drainage.
**Appendix A: Statistical significance of inter-city analysis**
The *NDVImax* data in and around the cities are very variable. In order to estimate the statistical
significance of the difference of multiple time series, we used a conservative hypothesis testing
with the Student's *t*-criterion. In this analysis, two statistical parameters are different at the
confidence level 95% ($\alpha = 1.96$, the light gray shading in Figs. 3 and 4) and 99% ($\alpha = 2.58$,
the dark gray shading) assuming the normal distribution of the difference errors when
$$\sqrt{\frac{2}{N-1}} \max_{m=1\ldots28} \left( \max_{k=0\ldots8} \sigma_n(k) \right) > \alpha \qquad (A1)$$
where $\sigma_n(k)$ is the standard deviation of a parameter in testing for the ring $k$ and the city $n$. We
want to stress that the conservative estimation is related to the differences, which have
significantly larger magnitude than the variations of *NDVImax* within each ring. Thus, the
differences appear to be more significant than the trends for each of the rings. There were found
no significant trends at levels higher than 90% for any 5-km ring around any city.



# 1 Acknowledgements

This study was supported by (1) the Belmont Forum and the Norwegian Research Council grant
HIARC: Anthropogenic Heat Islands in the Arctic: Windows to the Future of the Regional
Climates, Ecosystems, and Societies (no. 247268), (2) the Belmont Forum and the U.S.
National Science Foundation grant Collaborative Research: HIARC: Anthropogenic Heat
Islands in the Arctic: Windows to the Future of the Regional Climates, Ecosystems, and
Societies (no. 1535845), and (3) the Centre for Climate Dynamics at the Bjerknes Centre grant
BASIC: Boundary Layers in the Arctic Atmosphere, Seas and Ice Dynamics.



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



**Table 1**. List of 28 urban and industrial areas (cities) considered in this study. The city
population (pop. in thousands inhabitants) is estimated according to the Russian national census
for 2010. The mean background *NDVImax* (nature) is given for the most distant 40-km ring.
The relative trends are given for the time series without the years with the minimum and
maximum *NDVImax*. Statistically significant trends at 95% level are underlined. The biomes
are abbreviated as: tundra (T); forest-tundra (FT); norther taiga forest (NTF); and middle taiga
forest (MTF).

| N | City Name | Coord. | Pop. (x1000) | Biome | Mean NDVImax (nature) | NDVImax trend (nature) [% dec⁻¹] | Mean NDVImax (city core) | NDVImax trend (city core) [% dec⁻¹] |
|---|---|---|---|---|---|---|---|---|
| 1 | Beloyarsky | 63˚42'N 66˚40'E | ↓49 | NTF | 0.78 | -0.6% | 0.65 | +3.0% |
| 2 | Bovanenkovskiy | 70˚21'N 68˚32'E | 2 - 6 | T | 0.65 | +1.4% | 0.57 | **-10%** |
| 3 | Gubkinsky | 64˚26'N 76˚27'E | ↓26 | FT | 0.65 | -1.1% | 0.53 | +1.1% |
| 4 | Khanty-Mansyisk | 61˚48'N 69˚10'E | ↑93 | MTF | 0.78 | -2.6% | 0.62 | -3.0% |
| 5 | Kogalym | 62˚14'N 74˚32'E | ↑61 | MTF | 0.69 | -1.3% | 0.65 | -2.0% |
| 6 | Labytnangi | 66˚39'N 66˚25'E | ↑26 | FT | 0.75 | -2.0% | 0.60 | **+6.0%** |
| 7 | Langepas | 61˚15'N 75˚10'E | ↑43 | MTF | 0.77 | -1.9% | 0.72 | +1.4% |
| 8 | Megion | 61˚22'N 76˚06'E | ↓49 | MTF | 0.77 | +0.8% | 0.52 | **+6.9%** |
| 9 | Muravlenko | 63˚47'N 74˚31'E | ↓33 | NTF | 0.65 | -1.0% | 0.47 | -3.6% |
| 10 | Nadym | 65˚32'N 72˚31'E | ↓46 | NTF | 0.71 | +1.2% | 0.55 | -1.8% |
| 11 | Nefteugansk | 61˚55'N 72˚36'E | ↓126 | MTF | 0.76 | -2.1% | 0.56 | -3.0% |
| 12 | Nizshnvartovsk | 60˚56'N 76˚33'E | ↓266 | MTF | 0.78 | -1.8% | 0.60 | **-4.7%** |
| 13 | Novyi Urengoy | 66˚05'N 76˚04'E | ↓116 | NTF | 0.63 | **+4.0%** | 0.54 | +3.8% |
| 14 | Noyabrsk | 63˚11'N 75˚27'E | ↓107 | MTF | 0.69 | No change | 0.60 | +0.6% |
| 15 | Nyagan | 62˚08'N 65˚24'E | ↑56 | MTF | 0.79 | -0.7% | 0.59 | **+4.6%** |
| 16 | Pangody | 65˚51'N 74˚31'E | ↑11 | FT | 0.68 | +3.6% | 0.49 | -2.1% |
| 17 | Pokachy | 61˚44'N 75˚35'E | ↑17 | MTF | 0.74 | -1.5% | 0.53 | +1.6% |
| 18 | Purpe | 64˚29'N 76˚42'E | ↓10 | FT | 0.67 | -0.4% | 0.49 | **-4.3%** |
| 19 | Pyt-Yakh | 60˚44'N 72˚49'E | ↑41 | MTF | 0.81 | -2.0% | 0.67 | +1.1% |





| 20 | Raduzhnyi | 62°06'N 77°28'E | ↓43 | MTF | 0.72 | -1.5% | 0.65 | -0.3% |
|---|---|---|---|---|---|---|---|---|
| 21 | Salekhard | 66°31'N 66°36'E | ↑48 | FT | 0.75 | -0.9% | 0.54 | **+5.5%** |
| 22 | Sovetskiy | 61°21'N 63°34'E | ↑28 | MTF | 0.79 | -0.6% | 0.57 | 0.9% |
| 23 | Surgut | 61°15'N 73°24'E | ↑332 | MTF | 0.75 | -2.4% | 0.53 | -0.2% |
| 24 | Tarko-Sale | 64°55'N 77°47'E | ↑21 | FT | 0.69 | -0.6% | 0.51 | **-5.9%** |
| 25 | Tazovskiy | 67°28'N 78°42'E | ↓7 | NTF | 0.73 | +1.5% | 0.63 | **+7.6%** |
| 26 | Uray | 60°07'N 64°46'E | ↑**40** | MTF | 0.76 | -0.2% | 0.67 | +0.5% |
| 27 | Urengoy | 65°57'N 78°21'E | ↓11 | FT | 0.68 | -1.0% | 0.55 | -2.1% |
| 28 | Yar-Sale | 66°52'N 70°49'E | ↓7 | T | 0.66 | +3.9% | 0.52 | -2.8% |

2    **Table 2**. Fractional areas of the *NDVImax* changes for 2000–2014 in four NWS biomes

| *NDVImax* change | | Tundra | Forest Tundra | Northern Taiga | Middle Taiga |
|---|---|---|---|---|---|
| Highly negative | ≤ −0.006 | 0.28% | 0.18% | 0.73% | 1.05% |
| Slightly negative | −0.006 to −0.003 | 8.58% | 3.12% | 13.48% | 20.25% |
| Near zero | −0.003 to 0.003 | 23.56% | 15.01% | 39.19% | 43.18% |
| Slightly positive | 0.003 to 0.006 | 34.25% | 20.33% | 21.38% | 13.16% |
| Highly positive | 0.006 < | 33.33% | 61.36% | 25.22% | 22.36% |



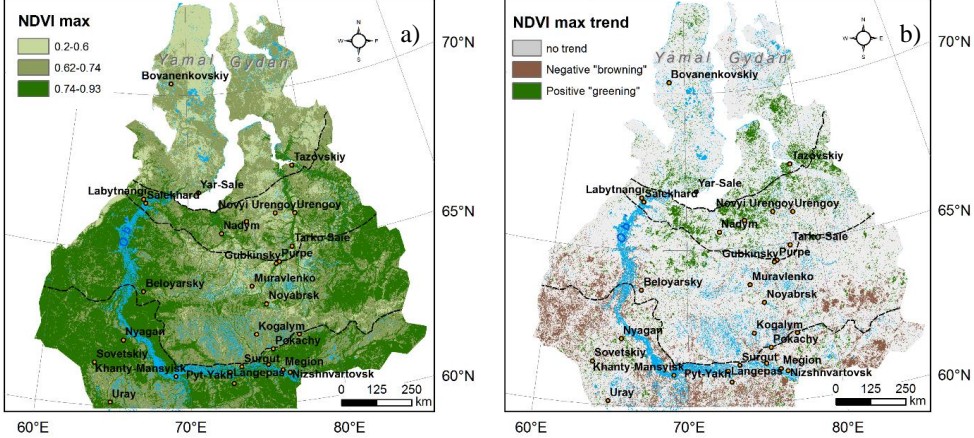

1    **Figure 1**. The 15-year mean absolute *NDVImax* (a) and the statistically significant (at $\alpha < 0.05$)

2    *NDVImax* trends for 2000-2014 (b).





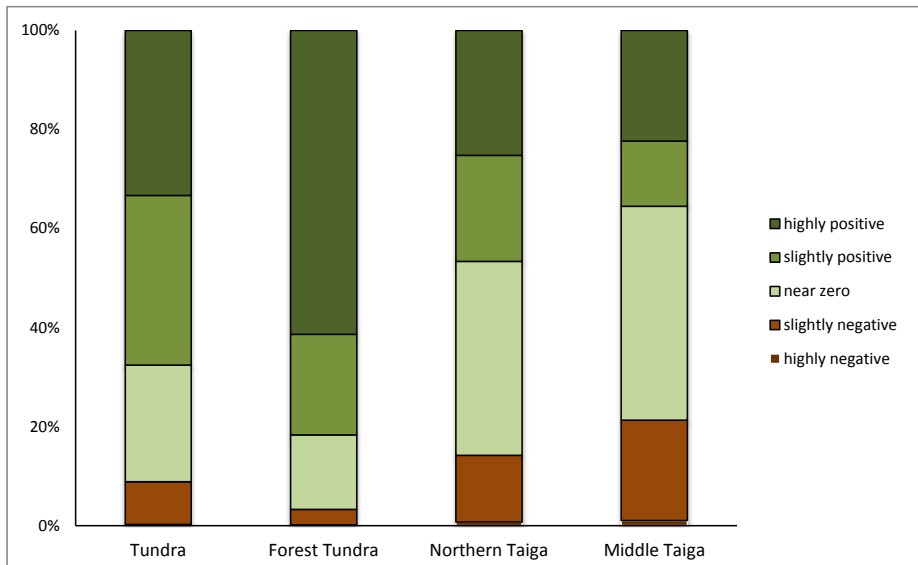

1    **Figure 2**. Total fractional areas of *NDVImax* changes for classes as defined in Table 2.













**Figure 3**. The statistical structure of *NDVImax*, correlations and trends at the city-core and the
8 rings in the buffer zones of Bovanenkovskiy (tundra), Nadym (tundra-forest), Noyabrsk
(northern taiga) and Surgut (middle taiga). The distance is given in km from the central pixel
of each city core. The upper panels for each city show the decay of correlations (extreme years
were excluded from the calculation) between the *NDVImax* variations at the city core and in the
corresponding distance. The lower panels show: bold line with squares – the mean *NDVImax*;
dark gray rectangle – one standard deviation of *NDVImax* for each ring; light gray rectangles –
three standard deviations; triangles – the years with the maximum (upward-looking triangle)
and the minimum (downward-looking triangle) of *NDVImax*; vertical black line with white
circle – the magnitude of the *NDVImax* change obtained as the trend multiplied by 15 years;
black circle – the same as the white circle but for the trends obtained when the maximum and
the minimum *NDVImax* were excluded.





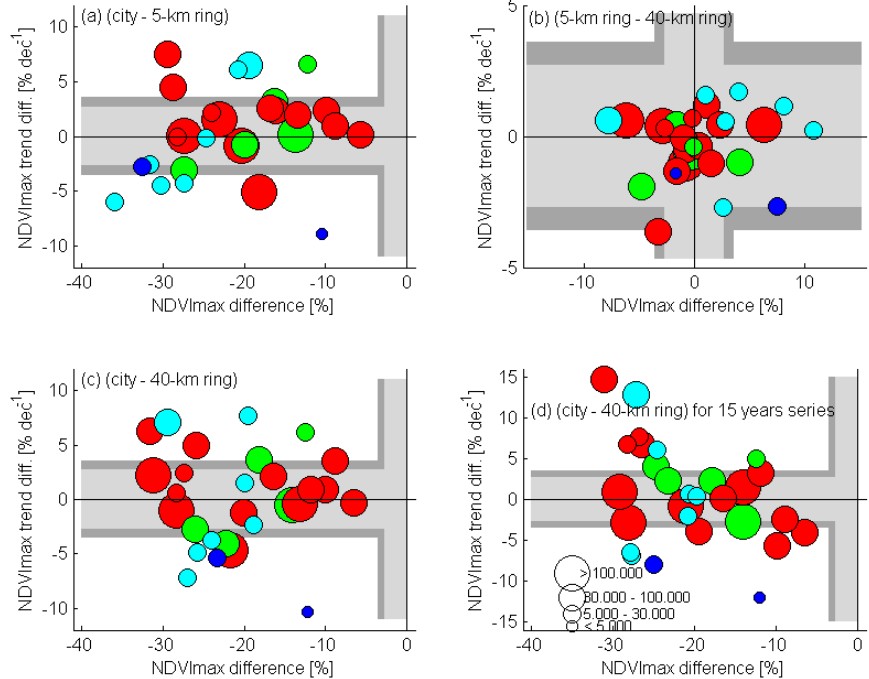

**Figure 4**. The relative changes $\Delta N(i,j)$ (x-axis) and $\Delta R(i,j)$ (y-axis) given in % for (a) the city core ($i = 0$) and the closest 5-km ring ($j = 1$); (b) the closest ($i = 1$) and the most distant background ($j = 8$) rings; and (c) the city core ($i = 0$) and the ring ($j = 8$). The numbers were obtained for the time series without the years of the maximum and minimum *NDVImax*. Panel (d) shows the same as (c) but for the full 15-year time series. The circle color indicates: blue – tundra; cyan – forest-tundra; green – northern taiga; red – middle taiga. The circle size indicates the city population in 2005 as shown on panel (d). The gray shading shows the statistical confidence envelopes (99% – dark gray shading; 95% – light gray shading).



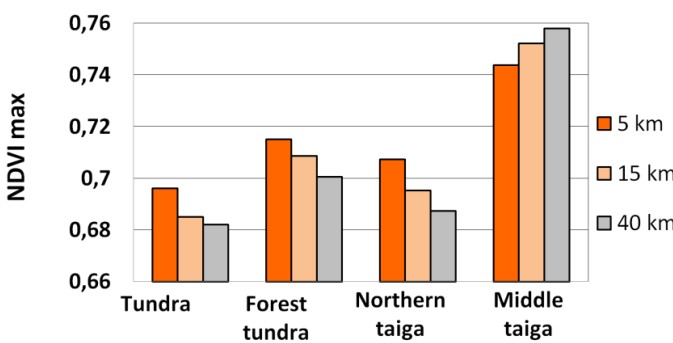

**Figure 5**. Aggregated *NDVImax* for 5-km (closest to the city core), 15-km and 40-km (the most

distant background) rings around the cities in the corresponding four biomes.

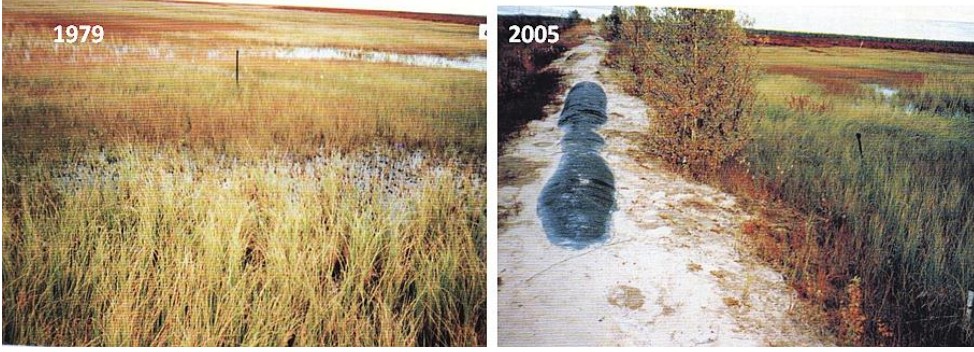

**Figure 6**. Illustration of the alternative ecosystem development on anthropogenically disturbed

patches. The vegetation cover changes along the gas pipeline Nadym-Punga.

