# Peer review of "Trends in normalized difference vegetation index (NDVI)"

_Atmospheric Chemistry and Physics, 2016_

## Referee Comment (RC1) · Anonymous Referee #1 · 31 May 2016

The main goal of the paper "Trends in normalized difference vegetation index (NDVI) associated with urban development in northern West Siberia" is to investigate an impact of urban development on vegetation trends in ifferents parts of the region . And I would like to note that this task is very well accomplished . Certainly, the main idea of article is clear, the title and abstract are appropriate and correspond to the content of the manuscript. The article topic is within the scope of a journal. I recommend to publish the article after a little technical corrections
* * *

---

## Referee Comment (RC2) · A. Colpaert (Referee) · 3 Jun 2016

Page 9 / 14, vegetation period, this is usually referred to as "growing season".

page 11, line 19-20 soil heat fluxes are quite complex, and I would say the better drainage just gives better rooting conditions for woody species.

It could be fruitful to discuss reasons for the greening / browning, what is the relative role of climate and what is the role of new infrastructures.

---

## Author Comment (AC1) · 6 Jun 2016

We thank the Referee #1 for appreciation of our work. Unfortunately, we are not sure what "a little technical corrections" are required. We will introduce the corrections when they will be specified.
* * *

---

## Author Comment (AC2) · 7 Jun 2016

1. We corrected "vegetation period" to "growing season" throughout the text.

2. We agree that better drainage is one of the key factors for establishing of the woody vegetation. We can illustrate this with shrub colonization along the winter trails near Nefteugansk (61ЁŽ55'18" N, 72ЁŽ36'11" E) (Landsat image), see the attached file as Fig. 1. This site is found in the middle taiga zone so that the soil temperature does not limit the shrub growth. At the northern treeline however the cause and result puzzle is more complex. Rooting conditions, soil moisture and water availability, soil thermal conductivity, mineralization and surface heat balance all found to be significant factors. We rewrite the corresponding paragraph on the page 11 to include three new studies

into the discussion, namely, Lloyd et al. (2003), Frost et al. (2013), and very recent study by Juscak et al. (2016). The conclusion of this discussion supports our points drawn from area-averaged satellite data analysis. It states that the availability of the disturbed microsites could be the limiting factor for the woody vegetation proliferation. Urbanization and subsequent mechanical disturbances and fairs create large number of such disturbances, particularly in 5 km buffer zone, which explains shrubification of this zone. However, in order to explain higher NDVImax, presence and growth of tall shrubs and trees are not enough as the field studies suggested. Warmer soils and generally higher air temperature is needed as it is follows from the rest of the observed literature.

The revisited manuscript with Review trace markup is attached as Supplement.

3. The manuscript to some degree provides the discussion on the relative roles of climate change and new infrastructure development (see Introduction, pages 2-3; and Discussion, page 10). In particular, we use the results of Macias-Fauria et al. (2012) and Bhat et al. (2013). However, we acknowledge that at present we cannot quantify the relative role of climate versus the urbanization impact. Therefore, we set to run another study with the aim to attribute the greening/browning to dominant species in the area.

Please also note the supplement to this comment:
http://www.atmos-chem-phys-discuss.net/acp-2016-51/acp-2016-51-AC2-supplement.pdf
* * *
[Figure]

Illustration of vegetation type change along the winter trails near Nefteugansk (61°55'18" N, 72°36'11" E)

**Fig. 1.**

**Supplement:**

[revised manuscript text omitted]